# Inverting COSMIC-2 Phase Data to Bending Angle and Refractivity Profiles Using the Full Spectrum Inversion Method

**Loknath Adhikari** [1,*]**, Shu-Peng Ho** [2] **and Xinjia Zhou** [3]

1 Cooperative Institute for Satellite Earth System Studies (CISESS), University of Maryland, College Park, MD 20742, USA

2 Center for Satellite Applications & Research (STAR), NESDIS/NOAA, College Park, MD 20740, USA; shu-peng.ho@noaa.gov

3 Global Science & Technology, Inc., 7855 Walker Drive, Suite 200, Greenbelt, MD 20770, USA; xinjia.zhou@noaa.gov

\* Correspondence: loknath@umd.edu

**Abstract:** The radio occultation technique provides stable atmospheric measurements that can work as a benchmark for calibrating and validating satellite-sounding data. Launched on 25 June 2019, the Constellation Observing System for Meteorology, Ionosphere, and Climate 2 and Formosa Satellite Mission 7 (COSMIC-2/FORMOSAT-7) are expected to produce about 5000 high-quality RO observations daily over the tropics and subtropics. COSMIC-2 constellation consists of 6 Low Earth Orbit (LEO) satellites in 24° inclination orbits at 720 km altitude and distributed mainly between 45°N to 45°S. The COSMIC-2 observations have uniform temporal coverage between 30°N to 30°S. This paper presents an independent inversion algorithm to invert COSMIC-2 geometry and phase data to bending angle and refractivity. We also investigate the quality of Global Navigation Satellite System (GNSS) and LEO position vectors derived from the UCAR COSMIC Data Analysis and Archive Center (CDAAC). The GNSS and LEO position vectors are stable with LEO position variations < 1.4 mm/s. The signal-to-noise ratio (SNR) on the L1 band ranges from 300–2600 *v/v* with a mean of 1600 *v/v*. The inversion algorithm developed at NOAA Center for Satellite Applications and Research (STAR) uses the Full Spectrum Inversion (FSI) method to invert COSMIC-2 geometry and phase data to bending angle and refractivity profiles. The STAR COSMIC-2 bending angle and refractivity profiles are compared with in situ radiosonde, the current COSMIC-2 products derived from CDAAC, and the collocated European Center for Medium-Range Weather Forecasts (ECMWF) climate reanalysis data ERA5. The mean bias at 8–40 km altitude among the UCAR, ERA5, and STAR is <0.1% for both bending and refractivity, with a standard deviation in the range of 1.4–2.3 and 0.9–1.1% for bending angles refractivity, respectively. In the lowest 2 km, the RO bias relative to ERA-5 shows a strong latitudinal and SNR dependence.

**Keywords:** radio occultation; Global Navigation Satellite System; excess phase; bending angle; refractivity; Full Spectrum Inversion



## 1. Introduction

Radio occultation (RO) techniques measure the time delay of carrier radio signals passing through the atmosphere between the Global Navigation Satellite System (GNSS) satellite and the receiver onboard low earth orbit (LEO) satellite [1]. Since the launch of the Constellation Observing System for Meteorology, Ionosphere, and Climate and Formosa Satellite Mission 3 (COSMIC hereafter) in 2006, RO data have been used in a wide range of applications in the atmospheric science community [2]. COSMIC RO data have been used as anchor references to calibrate other satellite data [3] and identify the in-situ radiosonde temperature biases [4]. Refs. [5–17] summarized COSMIC science applications from 2006 to 2019.

COSMIC has provided more than seven million RO soundings since its launch [2]. Recently, a COSMIC follow-up mission, the Constellation Observing System for Meteorology, Ionosphere, and Climate 2 and Formosa Satellite Mission 7 (hereafter, COSMIC-2), was launched on 25 June 2019. Using the TriG (Global Positioning System—GPS, GALILEO, and GLObal NAvigation Satellite System—GLONASS) GNSS-RO Receiver System (TGRS), COSMIC-2 can receive radio signals from the GPS, GLONASS, and Galileo. With six receivers onboard low inclination orbit, COSMIC-2 is expected to produce ~5000 RO measurements over the tropics and subtropics daily [5,18,19].

To obtain the atmospheric bending angle and refractivity profiles, we must first perform accurate orbit determination and phase calculations. One can determine GNSS and LEO satellite positions and velocities using the precise orbit determination (POD) method [20,21]. We can derive the excess phase using precise satellite orbit positions and velocities and corrected GNSS and LEO clock measurements [22,23]. When the accurate occultation geometry and excess phase are determined, we can retrieve bending angle (BA) and refractivity profiles. The uncertainty of RO retrievals depends on the (1) accurate calculations of the excess phase and (2) the retrieval approaches used to retrieve bending angle and refractivity profile (see Section 3). Ref. [24] has stated that different inversion approaches and implementation of initial conditions may contribute to retrieval uncertainties.

In the upper troposphere and above, where there are no sharp refractivity gradients, the RO signals are monotonic (i.e., each signal ray received at the receiver has a unique Doppler shift). The Doppler shift caused by the atmospheric effects is defined as excess Doppler. The excess Doppler can be determined by the derivative of the excess phase's time series. We can then convert the excess Doppler and satellite positions and velocities to the bending angle profile [1,25–27]. The approach to inverting geometry and phase data to bending angle profile is called the geometric optics (GO) inversion method [24–26]. However, in the presence of large refractivity gradients, multiple rays can arrive at the receiver simultaneously. Each of these rays has its Doppler shift. This phenomenon is called 'multipath' which results in significant uncertainties in the excess phase computation. Since multiple rays are present in the signal, the Doppler shift cannot be uniquely determined by differentiating the excess phase. As a result, the GO method usually fails to retrieve the bending angle as a monotonic function of the impact parameter. In the GNSS RO observations, multipath propagation occurs typically in the tropical and sub-tropical lower troposphere with large water vapor gradients.

Different approaches have been proposed to invert the RO data in the presence of multipath [28–35]. These approaches included the back-propagation method, radio-optics method, and Fresnel diffraction theory methods [30,32,33]. The wave optics (WO) method is a popular approach to handle the multipath issue. The WO method is based on Fourier operators, e.g., canonical transform (CT) and Full-Spectrum Inversion (FSI) [31,34]. Both CT and FSI methods use a global Fourier to transform the complex RO signal. The FSI method assumes that the orbit of the satellite is circular. However, even for short orbits for RO events, radial variations introduce significant errors and cannot be approximated as circular. Therefore, these radial variations have to be addressed for the FSI application. Although both CT and FSI reduce the complex RO signal to FFT, the CT method must first perform the back-propagation calculation, which is extremely computational-expensive. The phase-matching (PM) method [36] can also be applied to actual orbits without orbit corrections, but it does not reduce the RO signal to a single Fourier transform. The FSI method is one of the most computationally efficient methods to identify the individual frequencies present in the RO signal using the stationary phase approach [31,37].

The FSI method is commonly used for RO data processing, particularly in the lower troposphere. It is based on the assumption that the Fourier transform of the entire signal can be computed using the stationary phase method [31]. Ref. [31] demonstrated that for a uniformly circular orbital geometry, the Fourier transform phase derivative to frequency gives the central angle formed by the radius vectors to the LEO and the GNSS satellites,

and the total frequency of each ray is proportional to the impact parameter. As a result, we can use the impact parameter and the central angle combined with radius vectors to calculate the bending angle. We can then use Abel inversion to convert bending angle profiles to refractivity profiles [38].

One of the limitations of the FSI methods is that the global Fourier transform only works in a circular satellite orbit. Actual satellite orbits for a short duration of the RO period can be approximated as circular. However, RO signal inversion is based on the assumption that the atmosphere is spherically symmetric. The atmosphere can be approximated to be in spherical symmetry locally. Assuming that the horizontal variations are small, then transforming the coordinates relative to a local center of curvature as demonstrated by Syndergaard [39]. This coordinate transform causes the orbits to deviate from circular. Note that the circular orbit requirement can be approximated by correcting the radial variations in the occultation geometry. The radial variations caused by noncircular orbits can then be estimated by projecting the occultation positions to a predetermined circular orbit about the local center of curvature [31,40].

Different processing centers use different processing methods and noise filtering schemes that can cause structural uncertainty among RO products provided by other processing centers. An independently derived BA and refractivity profiles are beneficial for quantifying the processing inversion method-dependent structural uncertainty of the RO data products. In this study, we apply the FSI method to develop an independent RO processing package to invert COSMIC-2 excess phase to bending angle and refractivity profiles. We also used in situ and other data sets to assess the quality of our COSMIC-2 derived variables.

This study describes an independent processing system based on the FSI method developed at the National Environmental Satellite, Data, and Information Service (NESDIS) Center for Satellite Applications and Research (STAR). Based on the FSI method, we convert COSMIC-2 occultation geometry and phase data into bending angle and refractivity profiles (Figure 1). In this study, we first examine the quality of COSMIC-2 position vectors, the excess phases on L1 and L2 (i.e., L1b data), and their Signal Noise Ratio (SNR) characteristics. This is to ensure the quality of input data for our inversion algorithms. One month of COSMIC-2 L1b data is processed by using the FSI package. We first compare the STAR-derived refractivity profiles with those collocated in situ radiosonde measurements. The radiosonde temperature and moisture profiles are first converted to the refractivity profiles (see Section 2.3). We also validate the STAR FSI COSMIC-2 data products by comparing them with those parameters derived from the fifth-generation European Center for Medium-Range Weather Forecasts (ECMWF) Reanalysis (ERA-5). We also compare STAR COSMIC-2 bending angle and refractivity profiles to those derived from UCAR (University Corporation for Atmospheric Research) COSMIC Data Archive Center (CDAAC). UCAR COSMIC-2 data products have been validated by [18]. The specific UCAR COSMIC-2 data quality in terms of precision, stability, and accuracy are detailed in [18], which are no further described. Section 2 describes the data used in this study. The STAR FSI procedures are detailed in Section 3. We assess the quality of COSMIC-2 positioning and excess phase in Section 4. We evaluate the quality of STAR FSI data products in Section 5. We conclude our study in Section 6.

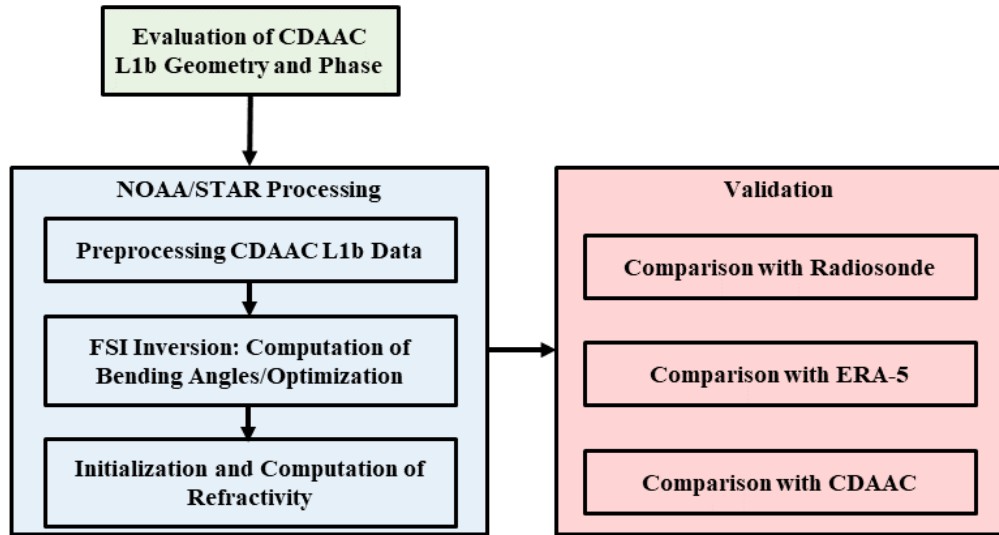

**Figure 1.** An outline of the NOAA/STAR data processing and validation scheme.

## 2. Data

*2.1. COSMIC-2*

COSMIC-2 constellation consists of 6 LEO satellites in 24° inclination orbits at a 720-km altitude. COSMIC-2 provides RO measurements within ±45° latitude region. The COSMIC-2 satellites are equipped with advanced receivers that can receive RO signals from GPS, Galileo, and GLONASS. At full deployment, COSMIC-2 can provide over 5000 high-quality RO measurements daily. Figure 2 shows the number of COSMIC-2 RO measurements since the early data acquisition period from 16 July 2019 (day of year DOY = 197) to 31 October 2019 (DOY = 304). Only GPS and GLONASS constellations are available in that period. In the earlier experimental phase of data acquisition (DOY 197–273), provisional data were acquired intermittently for testing purposes. As a result, there were days in which no data was received.

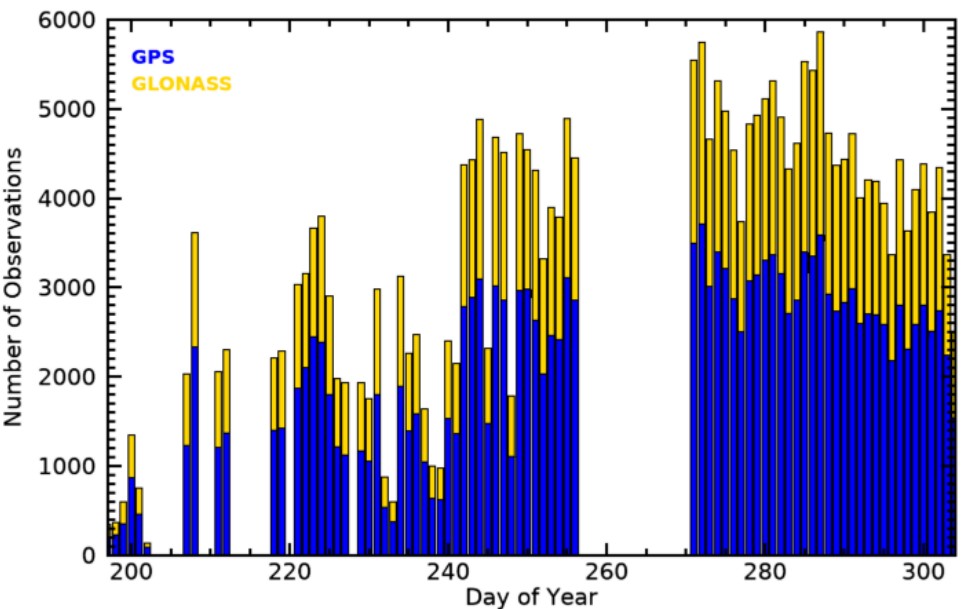

**Figure 2.** Daily count of COSMIC-2 RO observations from 16 July–31 October 2019. The count includes data obtained during the provisional data acquisition period.

Currently, UCAR CDAAC is the COSMIC-2 data processing center (DPC). CDAAC released the near-real-time (NRT) COSMIC-2 data for internal science team validation

after DOY 274 (1 October 2019). Note that not all COSMIC-2 satellites take measurements because of different technical requirements in the instrument deployment during the early period. Counts from individual satellites show that each satellite provides more than 900 RO profiles daily. When all six satellites are taking measurements, we can collect ~5000 occultations per day. The quality assessment of CDAAC COSMIC-2 products was summarized in [18].

With the low inclination orbit, the COSMIC-2 observations distribute more in the equatorial region and decrease poleward. Figure 3 shows the distribution of the COSMIC-2 observations for October 2019 (DOY 274–304). Figure 3a shows the observation count for that month at 5° × 5° longitude and latitude bins. The observation counts are larger in the tropics and decrease towards mid-latitudes. In the region of ±30° latitude, each 5° × 5° bin contains more than 100 RO measurements in October. Figure 3b depicts COSMIC-2 observations are relatively uniformly distributed over all local times from 30°N to 30°S. A uniform temporal distribution is essential to avoid temporal sampling biases in the observations, which is critically important for climate studies.

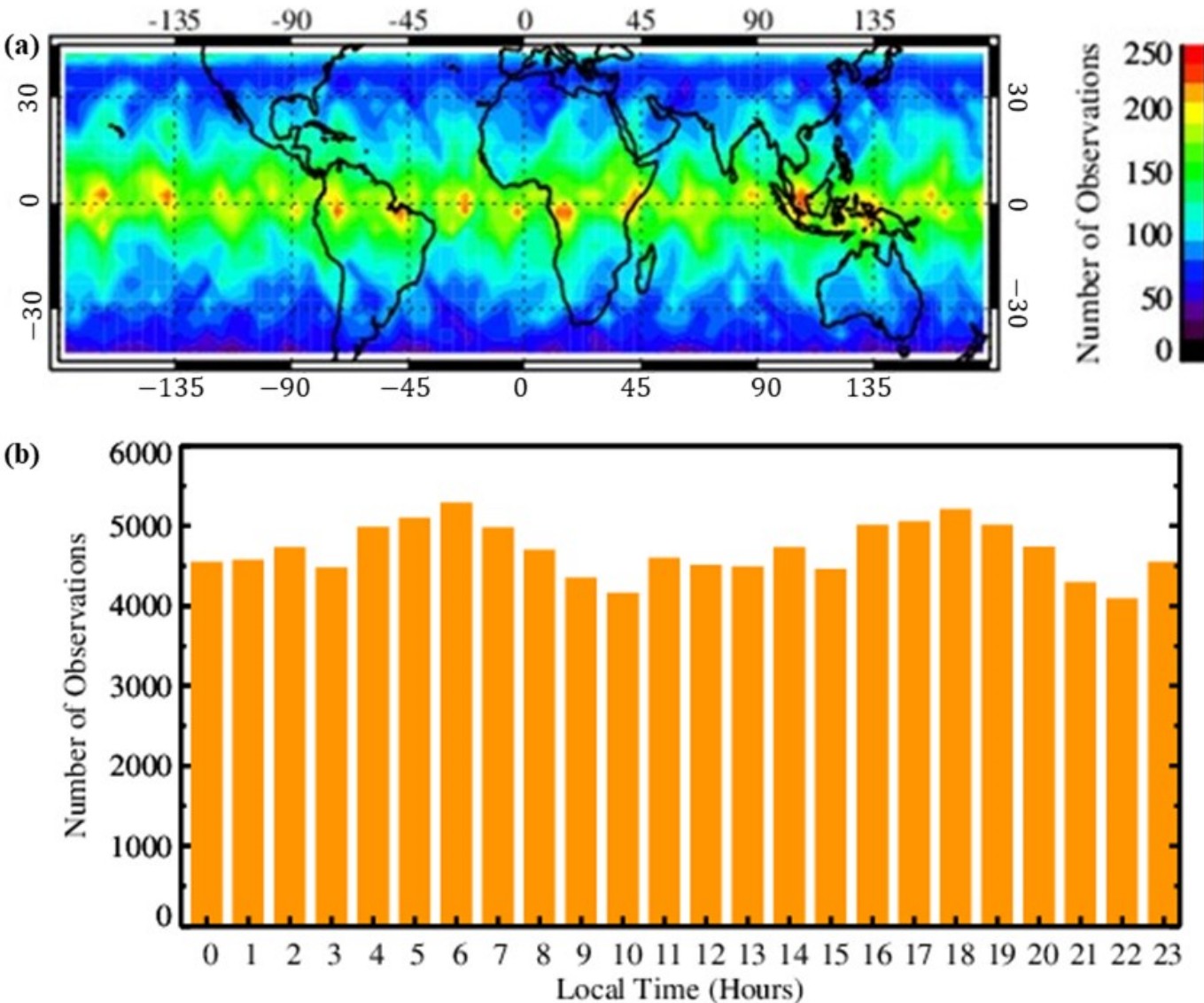

**Figure 3.** Distribution of COSMIC-2 observations (**a**) in the region ±45° latitude at 5 × 5° longitude and latitude bins and (**b**) at local solar times for the region ±30° latitude using observations from 1–31 October 2019.

This study uses COSMIC-2 L1b from 01 to 31 October 2019, as inputs for our FSI package. The COSMIC-2 L1b data are downloaded from UCAR CDAAC accessed on 9 March 2020 (https://cdaac-www.cosmic.ucar.edu/). The Level L1b (conPhs) files provide the information of occultation positions and velocities in Earth-centered inertial coordinate system (ECI), phase measurements, and signal to noise ratios (SNR) for two GNSS channels (i.e., L1 and L2 bands). We also downloaded CDAAC COSMIC-2 level 2 data (i.e., bending angle and refractivity profiles) to inter-compare with those derived from the STAR FSI package. The UCAR COSMIC-2 Level 2 refractivity data are derived using the GO method above 20 km and wave optics method phase matching (FM) below 20 km [23]. For the GO method, UCAR uses a 3-pass Savitzky-Golay filter (sliding polynomial regression) of the third degree and first order (to obtain derivative) and a 0.5-s window for the L1 excess phase and uses an unfiltered L1 excess phase for the FM method [23]. The two separated refractivity profiles are then sewed together to form a single profile [23]. In the current processing, we use the FSI method for the complete profile instead of using the GO and radio-holographic method applied in the UCAR processing. COSMIC-2 Level 2 data have been validated for their stability, precision, accuracy, e.g., [18]. This study compares the CDAAC COSMIC-2 BA and refractivity profiles to those derived from the STAR FSI package. We will discuss the causes of the difference between these two datasets.

*2.2. ERA5*

This study compares FSI BA and refractivity profiles with those computed from the ERA-5 (European Centre for Medium-Range Weather Forecasts-ECMWF reanalysis version 5). The ERA-5 is the most recent global atmospheric reanalysis data produced by the ECMWF [41]. ERA-5 has 31 km horizontal resolutions at 137 model level and contains advanced physics and data assimilation schemes with more observations spanning from 1979 than its predecessor ERA-Interim [41]. The ERA-5 provides error estimates from the 10-member ensemble of data assimilations at 63-km resolution. The current study uses the temperature ($T$), pressure ($P$), and specific humidity ($q$) at $0.25° \times 0.25°$ horizontal resolution and a 6-hourly temporal resolution with 37 pressure coordinate vertical levels. ERA-5 data are used to compare to STAR FSI products for both lands and oceans with its global coverage.

To prepare ERA-5 data to compare to those from FSI COSMIC-2 data, we apply the following procedures:

First, the ERA-5 $T$, $P$, and $q$ profiles are collocated in both space and time to COSMIC-2 reference tangent point location and time. This collocation in space and time allows a unique set of $T$, $P$, and $q$ profiles for each COSMIC-2 profile.

Second, using ERA-5 temperature and moisture profiles, we compute the ERA-5 refractivity as a function of altitude ($h$) is calculated using the following equation [1]:

$$N = 77.6\frac{P}{T} + 3.73 \times 10^5 \frac{e}{T^2} \tag{1}$$

where $N = (n - 1) \times 10^6$, $n$ is the refractive index, and $e$ is the water vapor pressure that can be obtained from specific humidity, $q$. The refractivity profiles are then interpolated to uniform 100 m vertical bins.

Finally, using the reference radius of curvature ($r_0$) of the corresponding COSMIC-2 profile, the ERA-5 bending angle ($\alpha$) profile is calculated as a function of the impact parameter ($a$) using Abel integration of the interpolated refractivity profile as [1,25]:

$$\alpha(a) = 2a \int_{r1}^{\infty} \frac{a}{\sqrt{n^2r^2 - a^2}} \frac{dln(n)}{dr} dr \tag{2}$$

and

$$r = r_0 + h \tag{3}$$

In our comparison, ERA-5 and RO profiles are interpolated to uniform vertical bins of 0.1 km from the surface up to 40 km. The fractional difference is calculated as a percentage difference ($100\% \times \frac{x_{star} - x_{era5}}{x_{era5}}$).

### 2.3. Vaisala RS92 Radiosonde

Radiosonde measurements provide in situ measurements of atmospheric variables. This study collocates available Vaisala RS92 radiosonde measurements with RO reference tangent point within the 300-km radius and 2-hour time window. From the pressure, temperature, and specific humidity measurements, we use Equation (1) to calculate the refractivity. We apply the following criteria to match RO-ROAB pairs: (1) select RO profiles that are flagged as 'good' and (2) remove COSMIC-2 profiles with L2P SNR < 300 *v/v*. A total of 620 radiosonde measurements are collocated to RO profiles for October 2019. The RO refractivity profiles are interpolated to the radiosonde heights for comparison.

### 3. NOAA STAR RO Data Inversion System Using FSI

References [23,42] illustrated that the significant steps for RO data processing include: (1) LEO POD and clock estimation, (2) determination of the excess phases and (3) retrieval of higher-order atmospheric products. This section details the STAR inversion package procedures that invert excess phase and amplitude to bending angle and refractivity profiles. Figure 4 shows the flowchart of the steps used in the inversion system. The description of each step is provided in the proceeding sub-sections. Table 1 shows the implementation approaches used in each step.

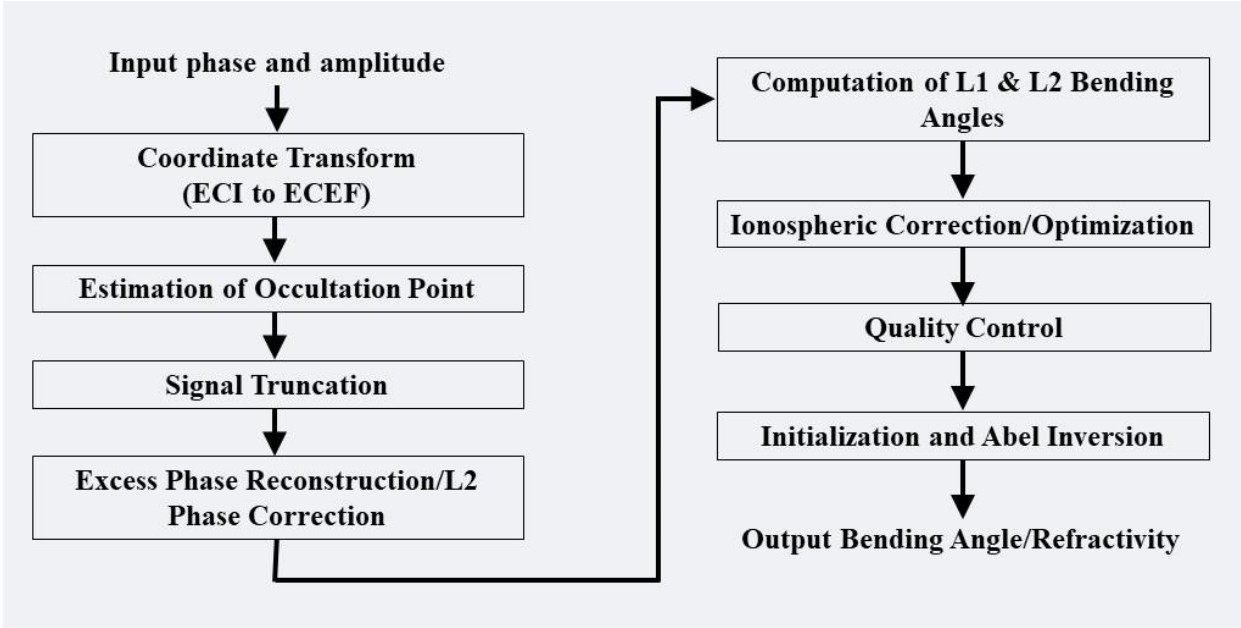

**Figure 4.** Flow chart depicting the different steps used in the NOAA STAR inversion of geometry and phase data to bending angle and refractivity profiles.

**Table 1.** Overview of the implementation of the STAR processing chain.

| Processing Step | Implementation Approaches |
|---|---|
| Input data | Input UCAR orbit in Cartesian ECI coordinates, L1 and L2 excess phase, and SNR data |
| Coordinate Transform | Transforming ECI coordinates to ECEF Coordinate |
| Signal Truncation | Based on L1 SNR, truncating signals using a threshold on calculated base SNR |

**Table 1.** *Cont.*

| Processing Step | Implementation Approaches |
| --- | --- |
| Excess Phase Reconstruction | Computation of excess phase after Fourier filtering of Doppler using 0.5-s window |
| Bending Angle Computation | Full-Spectrum Inversion |
| Ionospheric Correction | Linear combination and statistical optimization of L1 and L2 bending angles |
| Quality Control | Mean L1–L2 difference at 25–50 km < 100 μrad, mean fractional bending angle difference (COSMIC2-CIRAQ) at 25–40 km < 0.5 |
| Initialization | Exponential fit above 55 km |
| Refractivity Calculation | Abel inversion of the ionospheric corrected bending angle with the exponential fit |

### 3.1. Data Inputs

The input data for the inversion system is CDAAC provided Level L1b dataset. The Level 1b data contains the time series of GNSS and LEO satellite positions and velocities in ECI coordinate systems and the time series of excess phase and SNR in L1 and L2 bands. The SNR is a measure of the signal power at the receiver and is proportional to the signal amplitude normalized by estimated noise power. It is estimated calculated at the receiver and provided as an RO observable for the L1 and L2 bands. The excess phases have removed navigation data modulation (NDM). Therefore, unlike the COSMIC data, the current COSMIC-2 excess phase data does not require NDM removal in the excess phase pre-processing. Table 1 shows the implementation schemes of the STAR processing chain.

### 3.2. Convert L1 Data from Cartesian Earth-Centered Inertial (ECI) Coordinate to Earth-Center Earth-Fixed (ECEF) Coordinate

UCAR Level L1b geometry data are in the J2000 Cartesian Earth-centered Inertial (ECI) coordinate system. The first part of the processing involves converting the ECI coordinate system to the Earth-center Earth-Fixed (ECEF) coordinate system. The conversion includes determining the rotation matrix for the given ECI coordinate system to convert to the ECEF coordinate system. We use the International Earth Rotation and Reference System (IERS) convention 2010 to determine the rotation matrix [43,44]. This step is required so that the tangent points' position can be calculated at each timestamp to determine each profile's representative occultation point.

### 3.3. Transform the Reference Frame to the Local Center of Earth's Curvature

The inversion of the excess phase to bending angle is based on the assumption of a spherically symmetric atmosphere [24]. The spherical symmetry assumption requires satisfying two conditions: (1) there are no horizontal variations in the refractive index, and (2) the earth shape is perfectly spherical. The horizontal variations in the refractive index are less than 0.1%, so that the first approximation is generally a good approximation [1]. The phase data's inversion needs to be performed relative to the local center of curvature to satisfy the second approximation. The RO data is spherically symmetric close to the local center of curvature tangential to the Earth's ellipsoid [1,39]. In all the RO measurements, the tangent points drift with altitude, so a representative reference position needs to be determined to assign to each RO measurement. In the STAR processing, the curvature center is determined when the straight-line tangent point intersects the Earth's ellipsoid.

### 3.4. Determine the Signal Truncation Point

RO receivers take measurements in the open-loop (OL) mode record signals even after losing track of the occulting GNSS satellite [45,46]. This causes the receivers to record both the useful signal and the noise. Signal truncation determines and extracts the useful signal in the RO data taken in OL mode from the noise for processing. To determine the useful signal, first, the noise part of the data needs to be determined. We remove the noise part of

the data by first calculating the background SNR, then using an empirically determined threshold method to determine the point that separates useful data from noise. Signal truncation provides challenges in RO data processing because truncating the signal too high may result in loss of the higher bending angle components near the truncation point, causing negative bias in the retrieved bending angle and a decrease in the penetration depth. On the other hand, truncating the signal too low may add a noise-dominated signal that causes biases in the bending angle retrieval.

In the STAR processing, the truncation point is determined in two steps. (1) The background SNR is determined for each time series using the lowest 10-s of the data, smoothed using a coarse 3-s moving average. Starting from the lowest tangent point of the time series, a point is determined where the SNR is three times the background SNR. (2) Once the background SNR is defined, the final cutoff point is determined by going backward towards lower tangent points where the SNR first drops below 1.5 times the background SNR. These thresholds are subjective but necessary so that small SNR jumps observed at low levels in the tropical RO observations are excluded in the processed profile. Increasing the cutoff point thresholds causes a reduction in the penetration depth, and a decrease in the threshold causes a large part of noise to remain in the profile. Occurred more frequently over the tropical region and mid and high latitudes, RO profiles usually have spikes in the SNR profiles even after the receiver no longer receives signals from GNSS satellites.

Figure 5 shows a typical example of the signal's truncation for RO soundings taken on 1 October 2019 (C2E1.2019.274.01.19.G08). The original signal is in black, and the truncated signal is in red. Figure 5a shows the SNR, and Figure 5b shows the excess Doppler of the original/truncated signal. The figure illustrates that the SNR after 90 s reaches a background value. The excess Doppler shows that the significantly more significant variations are exceeding 10 m/s after the truncation point. This variation in excess Doppler indicates that the phase data constructed in the Level 1a to Level 1b processing is dominated by noise and should be discarded before inversion.

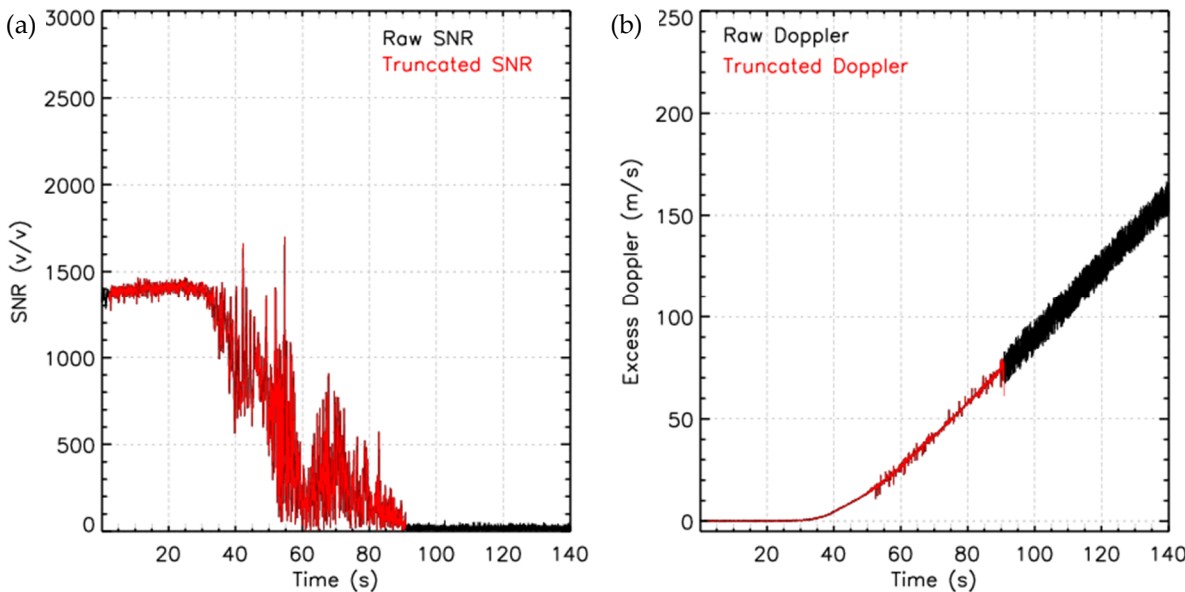

**Figure 5.** An example of the signal truncation scheme used in NOAA STAR processing, (**a**) SNR of the original/truncated signal in black/red and (**b**) excess Doppler of the original/truncated signal in black/red.

### 3.5. Noise Filtering of the Signal

The processing of the RO signals requires filtering out the noise present in the data that affects the retrieved bending angles [47–49]. The effect of noise can be reduced by applying a low-pass filter (e.g., [49]) or a radio-holographic filter (e.g., [48]). In the current version of

the inversion, we use a 0.5-s Fourier filtering of the excess Doppler. First, we computed the excess Doppler as the time derivative of the excess phase. Then, we apply the filter to the excess Doppler. The excess phase is then reconstructed from the smoothed excess Doppler. The 0.5-s smoothing window is selected so that the smoothing applied is close to the RO measurements' vertical resolution, approximately equal to the first Fresnel zone [1]. In the lowest 10-km impact height, the raw unsmoothed signal avoids removing small-scale atmospheric variations caused by rapidly varying refractivity structures. Figure 6a shows the Doppler for the raw signal and the smoothed signal for the RO measurement shown in Figure 5. Figure 6b shows the phase differences between the raw and the filtered signals at each time step of the 100 Hz COSMIC-2 signals.

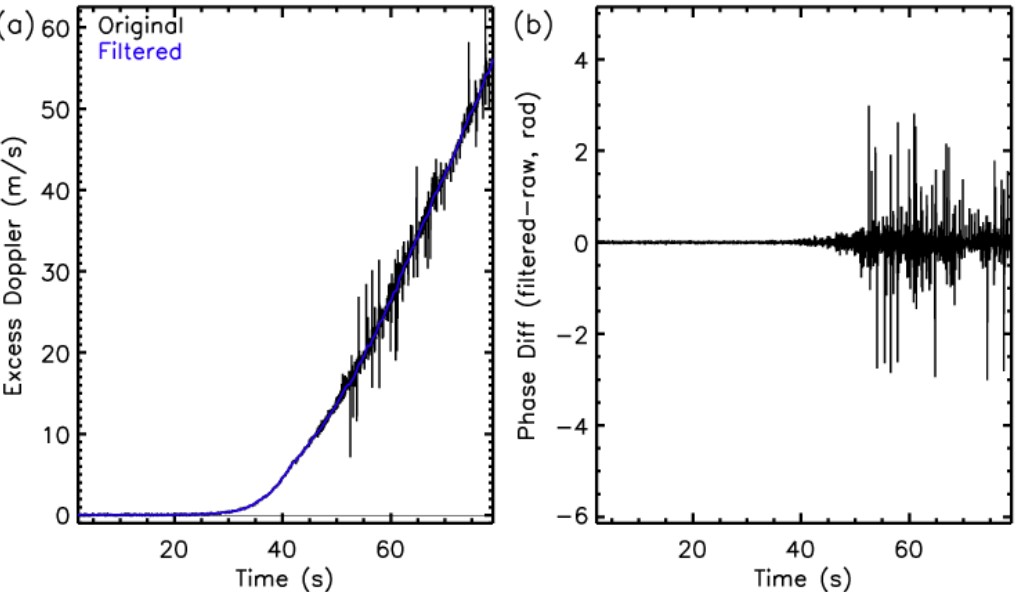

**Figure 6.** (**a**,**b**) Excess Doppler for the RO sounding on 1 October 2019.

### 3.6. Using the FSI Method to Determine L1 and L2 Bending Angles

The FSI method is used to invert the truncated RO data into bending angle profiles. Raw bending angles are retrieved for the L1 and L2 bands separately. For the L1 band, the smoothed signal is used above 10 km, and the raw signal is used below 10 km. For the L2 band, the smoothed signal is used for the whole profile. The 10 km boundary is selected for the use of the smoothed and raw L1 signal because the neutral tropospheric variations in the bending angles caused by the variability in water vapor content are confined to the lower troposphere, and the filtering of the lower tropospheric signal might affect the natural variation of the bending angle structure.

Because the bending angle profile inversion using the FSI method provides impact parameters and bending angle pairs for an infinite range of values for the impact parameter, it is crucial to determine the lowest impact parameter and bending angle pair. The current inversion of the STAR inversion uses the amplitude of the Fourier transform to specify the lowest impact parameter and bending angle pair. We first normalize the amplitude by amplifying the signal's mean amplitude between 10 and 50 km. The lowest point is then determined where the amplitude is less than 0.5 of the normalized amplitude.

Figure 7 shows the determination of the lowest impact parameter and the corresponding bending angle. Figure 7a shows the raw bending angle profile in blue and the cutoff profile in red. Figure 7b shows the normalized amplitude of the Fourier transform and the cutoff impact parameter. The FSI amplitude varies rapidly in the lowest part of the troposphere due to the rapidly fluctuating signal amplitude caused by different signal waves' superposition. At the end of the profile, the signal gradually drops to zero. The

figure illustrates the importance of the cutoff height's proper choice because the bending angle below the cutoff impact height (shown as the blue curve in Figure 7a) is purely noise.

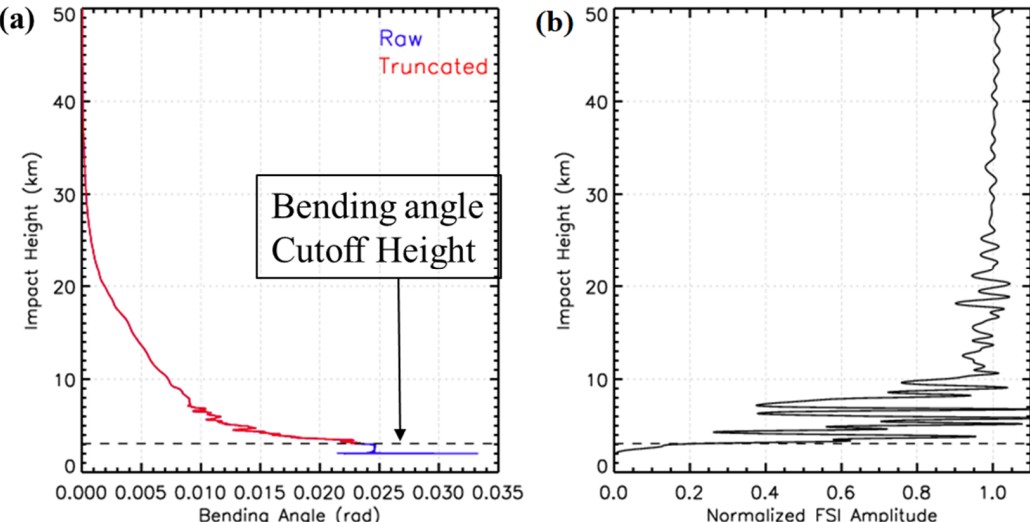

**Figure 7.** (**a**,**b**) Illustration of the impact parameter cutoff point in the FSI method.

### 3.7. Ionosphere Correction and Optimization of Bending Angle

We applied an ionospheric correction to remove the effects of the ionosphere on the neutral atmospheric profiles. Optimization of the bending angles is a method to reduce noise in the inverted bending angles. The ionospheric correction and optimization are conducted using Gorbunov's statistical optimization method [47]. The optimized bending angle ($\alpha(p)$) is expressed as [47]

$$\alpha(p) = \alpha_{BG(p)} + \frac{\sigma^S}{\sigma^S + \sigma^N}(\alpha_{LC}(p) - \alpha_{BG}(p) \tag{4}$$

where, $\sigma^S$ and $\sigma^N$ are covariances of the neutral atmospheric signal and residual errors, respectively, and $\alpha_{BG}$ is the background signal. $\alpha_{LC}$ is the linear combination of the bending angles at two channels used in RO measurements given by

$$\alpha_{LC}(p) = \frac{\alpha_1(p)f_1^2 - \alpha_2(p)f_2^2}{f_1^2 - f_2^2} \tag{5}$$

where, $f$ represents frequencies and subscripts *1* and *2* refer to the two RO channels.

We use the climatological model data CIRA86aQ_UoG (hereafter, CIRAQ) as the background references. Then, we compute the deviation ($\Delta\alpha$) of the L1 and L2 bending angles from the model bending angle ($\alpha_{L1,L2} - \alpha_m$). Then, we use the deviation to calculate covariance matrices for ionospheric and tropospheric signals and noise. For the ionospheric signal and noise covariance, $\Delta\alpha$ values between impact heights (impact parameter—a local center of curvature) between 50–70 km are used. For the neutral atmospheric signal covariance, the $\Delta\alpha$ values between 12–35 km are used. In the lower troposphere near the surface where the L2 signal is weak, the ionospheric correction is done by applying a constant ionospheric correction value calculated at the lowest impact height where L2 data is available.

### 3.8. Quality Control on the Inverted Bending Angles

The bending angles are inverted separately for the L1 and L2 bands and smoothed using a sliding window of 125-m. Using the retrieval from the L2 band, bad profiles are flagged using the lowest L2 impact height where all the profiles with the lowest L2 impact heights > 20 km are discarded because the ionospheric correction uncertainties will be

large in the absence of the L2 bending angle. The bending angle quality is determined for two different criteria: (1) the difference between the mean L2 and the L1 bending angle at 35–50 km impact height and (2) the mean difference between the corrected bending angle and model bending angle at 25–40 km. Profiles that pass both or fail either of the quality control criteria are assigned QC flags '0' and '1', respectively, where '1' represents a 'bad' profile flag. The quality determination using the first criteria flags the profile as 'bad' if the L2–L1 bending angle difference is >100 μrad. The 100 μrad thresholds are used because it is of the same order as the raw bending angle profiles in the L1 and L2 bands, and a difference > 100 μrad indicates an error in the retrieval of either the L1 or L2 bands. If the profile passes the criteria (1), then the second criteria are applied. The second quality control is used to optimize the stratosphere's optimized bending angle above tropospheric effects' altitudes. The profile is flagged as 'bad' if the mean difference between the optimized bending angle and climatological bending angle in the impact height range 25–40 km is greater than 50%. Since the altitude range selected for this QC criteria is above the tropopause, the variations between RO and climatology are small, and deviation of the RO bending angle > 50% of the climatological bending angle indicates errors in the RO bending angle.

### 3.9. Calculation of Refractivity Using Abel Integration

Refractivity is retrieved by Abel inversion [38]. We applied the Abel inversion on the ionospheric corrected bending angle to calculate the refractivity profile as a function of geometric altitude. For the refractivity calculations, the bending angle profiles up to 55 km are used. The bending angle above 55 km height is estimated by an exponential function with a scale height of 7 km.

## 4. Assessment of COSMIC-2 Positioning and Excess Phase

Before putting CDAAC's COSMIC-2 L1b data in the STAR FSI system, we first assess the stability of COSMIC-2's positions, the SNR, and the excess phase of the COSMIC-2. We also assess the quality of the UCAR processed COSMIC-2 bending angle profiles in this section.

### 4.1. Stability of Satellite Positions

We first examine the geometry stability of COSMIC-2 occultation, i.e., their position and velocity coordinate. The stability was calculated using the incremental velocity ($\Delta v$), which is the difference of the time derivative of the position vectors, where $\Delta v = v_{t+1} - v_t$.

Figure 8 shows an example of the incremental velocities for COSMIC-2 (Figure 8a–c) and GNSS (Figure 8d–f) for COSMIC-2 satellite flight module 1 (C2E1) and GPS satellite G03 at 00:21 Z on 9 October 2019. The daily mean absolute deviations (MAD), which are the daily means of the absolute incremental velocity for 15–27 October, are shown in Figure 9. An unstable satellite position is characterized by fluctuating values of the incremental velocity vectors. The figure shows that the incremental velocity vector variations are small (in the order of less than 0.1 mm/s), indicating the high stability of the positions for COSMIC-2. The daily mean absolute deviation of the incremental velocity shows that the daily averaged values for the GPS positions are less than 0.1 mm/s and COSMIC-2 is ~1.4 mm/s.



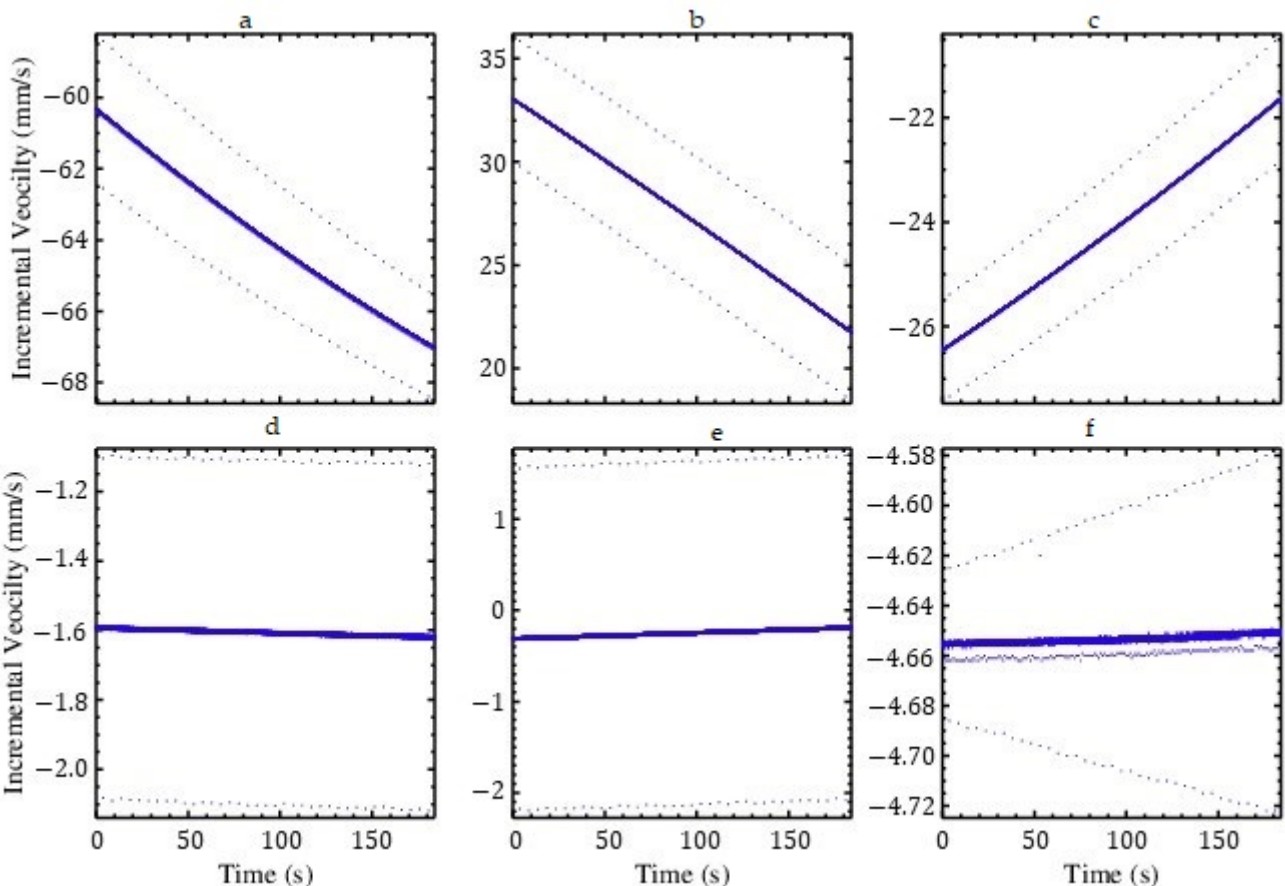

**Figure 8.** Incremental velocity vectors as a function of time for (**a**–**c**) COSMIC-2 and (**d**–**f**) GPS for the RO measurements at 00:21 Z on 9 October 2019.

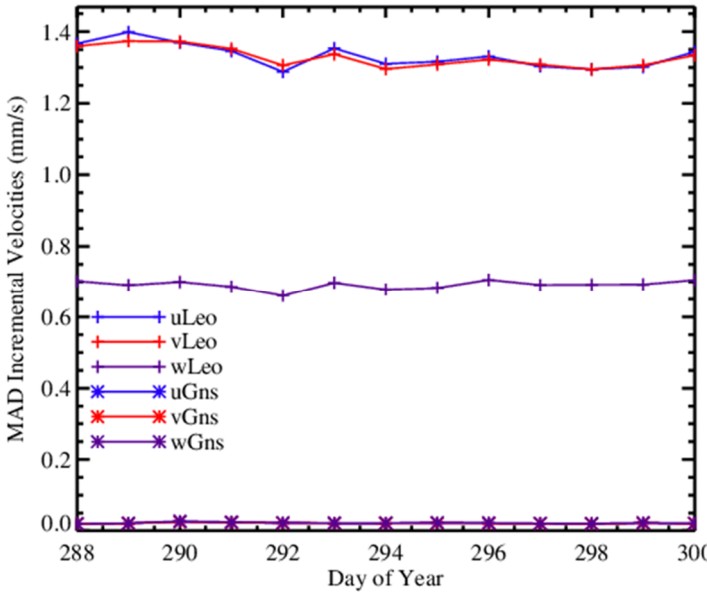

**Figure 9.** Mean absolute deviation (MAD) of the incremental velocity vectors for 15–27 October 2019.

### 4.2. Quality Assessment of Signal to Noise Ratio (SNR) and Excess Phase

Since the GNSS radio signals attenuation increases with the increase of the atmospheric density and water vapor pressure, the RO signal strength decreases as the signals traverse progressively lower to the earth's surface. Defocusing causes a superposition of the waves,

resulting in highly variable signal strength received by LEO receivers in the regions with rapidly varying refractivity gradients. SNR is an RO variable that represents signal strength at the receiver. As SNR decreases, the signal becomes progressively dominated by noise. It is difficult for measurements with relatively low SNR to distinguish actual signals from noise because noise becomes dominant signals. The height at which the SNR becomes small enough that the noise dominates the signal depends on the (1) signal power, (2) receiver noise, and (3) atmospheric conditions. A higher peak SNR generally means that the signal can penetrate lower into the atmosphere. Schreiner et al. [19] showed that the penetration depth of the COSMIC-2 profiles increases with increasing SNR. The COSMIC-2 receivers were designed to produce higher SNR than its predecessor, COSMIC (~800 *v/v*, not shown). Figure 10 shows SNR characteristics for RO measurements during October 2019. Figure 10a shows the daily mean SNR for October 2019. The mean SNR is computed from the 60–80 km impact height. The daily mean is calculated from the mean SNR for all the measured RO profiles for the day. Figure 10a shows that the daily mean SNR is in the range 1300–1500 and 500–650 *v/v* for the L1 and L2 bands, respectively. Since there are more than 3000 RO measurements per day, they represent the mean SNR for COSMIC-2 measurements. Figure 10b shows the relative frequency distribution of the L1 and L2 SNR taken at a bin size of 100 *v/v*. The L1 SNR varies from 300 to 2600 *v/v*, with the maxima at 1600 *v/v*. The L2 SNR has dual maxima at 500 and 1000 *v/v*. The two local maxima on L2 SNR are mainly due to different L2 SNR characteristics of the GPS and GLONASS satellites, as illustrated in reference [13].

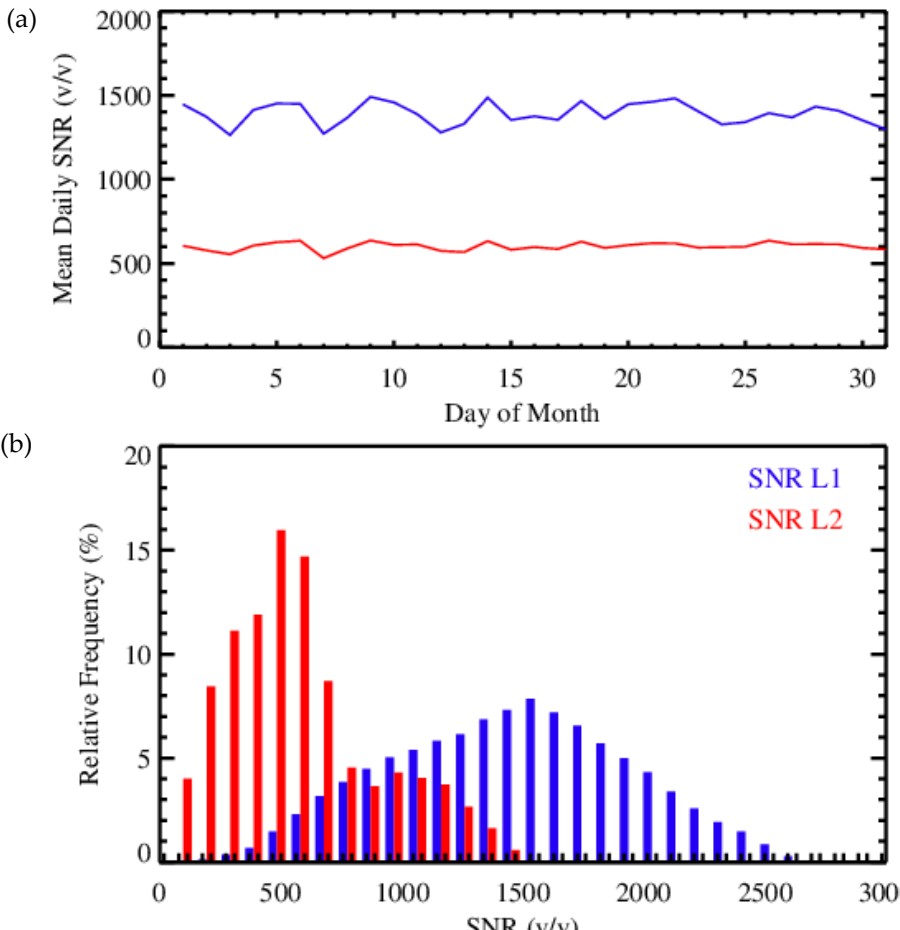

**Figure 10.** COSMIC-2 SNR characteristics: (**a**) the daily average SNR and (**b**) frequency distribution of SNR for October 2019.

The RO bending angle is derived from the RO signal's Doppler velocity. Since the Doppler velocity's atmospheric component is the derivative of the excess phase, the excess phase is the fundamental measurable in an RO event. Therefore, the RO bending angle's quality is directly affected by how the excess phase's noise is handled. In this section, the excess phase quality is determined by investigating Doppler velocity variation at consecutive time steps. The Doppler velocity for each occultation is determined at −20 km to 50 km straight-line impact height. This impact height is used because it represents the data where multipath is least likely to occur. The time derivative of the excess phase represents the atmospheric contribution to the Doppler velocity of the signal. The maximum difference of the Doppler velocity at each 1/10th second is calculated to represent the Doppler variation time series. Here, we use Doppler variation to represent the maximum variation in the Doppler velocity in 1/10th seconds of measurements. The variation is then averaged over for each profile.

The Doppler variation is a measure of the small-scale fluctuations in the excess phase measurements. Noisy and bad excess phase data usually have considerable Doppler variation. Figure 11 shows the Doppler variations for RO measurements on 19 July 2019, for L1 and L2 bands. Doppler variations of ~0.5 m/s represent natural variation caused by atmospheric effects and measurement noise at the selected altitude range. Figure 11 shows that ignoring the anomalous cases of bad excess phase data, which account for <1% of the full profiles, the average Doppler variation is close to 0.5 m/s within the range of atmospheric variations. The anomalous cases where the Doppler variations are much more significant than 0.5 m/s are flagged as 'bad' data and removed from the processing. However, in the current processing, we process all the available L1b data and provide a quality flag, as discussed in Section 3.8. Note that the L2 variations are more significant than L1 because L2 signals are weaker and do not penetrate as low as L1 signals. The L2 signal near −20 km impact height is noisier compared to the L1 signal.

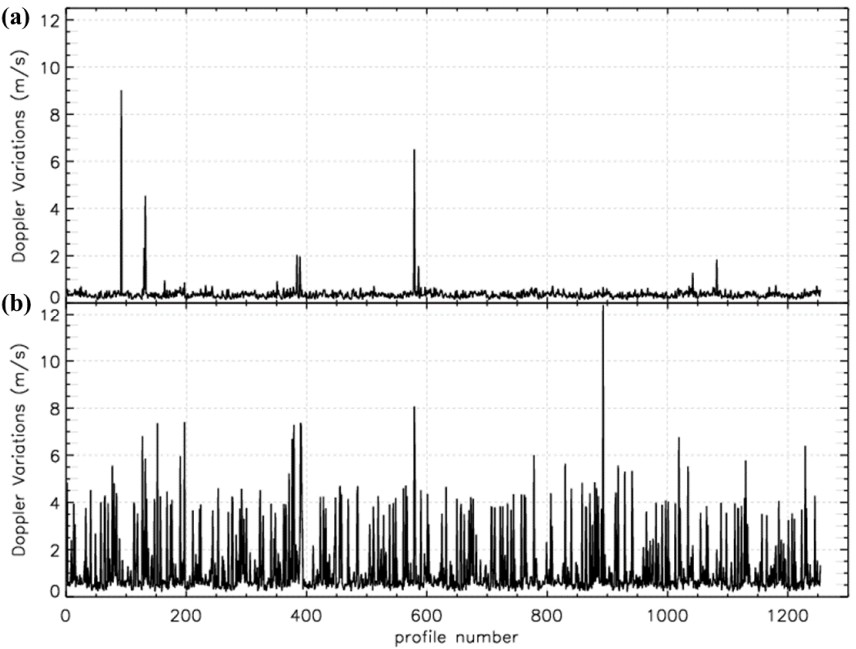

**Figure 11.** Variation of (**a**) L1 and (**b**) L2 Doppler for COSMIC-2 data on 19 July 2019.

## 5. Assessment of the Quality of the STAR FSI COSMIC-2 Data Products

To validate the STAR FSI RO products, we compare the STAR results with collocated radiosonde measurements (RAOB), reanalysis data (ERA-5), and UCAR processed COSMIC-2 products. It is a challenge to compare RO measurements and data products with in-situ observations and model outputs. RO data are derived from limb measurements, where its horizontal resolution is highly dependent on the atmospheric structure and its

variation with altitude. RAOB data are in-situ measurements and are sensitive to sensor response time. Similarly, reanalysis data are generated at specific time intervals and have limited vertical resolutions. Therefore, the RO products' comparisons with these different measurements and reanalysis products are limited by the differences among these data and their uncertainties. In this section, we present the comparison results among these datasets.

*5.1. Comparison to Radiosonde Observations*

The STAR FSI refractivity is compared to collocated radiosonde measurements in this section. Figure 12a,c show the mean fractional differences between these two datasets for different latitudinal bands and different SNR ranges, respectively. Their standard deviations are shown in Figure 12b,d, respectively. The sample number for the RO-RAOB pairs at the vertical levels are shown in thin lines. Black lines represent the comparisons for all latitude bands and SNR values. The number of RO-RAOB pairs used for the comparisons varies with height because not all radiosondes reach above 300 hPa (~16–18 km).

Figure 12a,c shows that the STAR refractivity has a positive bias in the lower troposphere above ~1 km and a slight negative bias in the upper troposphere and stratosphere. The mean and the standard deviations are 0.13% and 2.63%, respectively, for all the RO-RAOB pairs from the surface up to 25 km. The mean fractional difference is more significant in the lower troposphere below 5 km than those above 5 km, which shows more considerable variability among different latitudes and SNR bands.

Table 2 shows the mean and standard deviation of the fractional difference for altitudes below 5 km. There is a negative bias for mid-latitudes on both the hemispheres and below ~2 km for RO-RAOB pairs with SNRs < 1000 *v/v* and a positive bias for tropical and higher SNRs at altitudes below 5 km. The standard deviation shows a significant difference in the different latitude bands but shows that it increases for RO-RAOB pairs with <500 *v/v* and 1000–1500 *v/v*.

Two factors may cause the significant mean differences in the RO-ROAB in the lower troposphere:

(1) RO observations are from a wide horizontal scale, where radiosondes are from in-situ measurements, and

(2) the uncertainties in the RO bending angles are significant due to multiple frequencies at a given impact parameter. As a result, the difference between in situ and limb measurements can be large when the water vapor horizontal variation is large. In the lower troposphere, refractivity variations are highly dependent on water vapor variations because the magnitudes of water vapor variations are larger than for temperature and pressure.

Note that most of the RO-RAOB pairs are collect over lands and islands over the ocean. We further present the RO-ERA-5 comparison over both lands and oceans in Section 5.2.

**Table 2.** The mean (Standard Deviation) of the RO-ROAB fractional difference and the number of collocated profiles at different latitude bands and SNR ranges.

| Latitude Bands | STAR-ROAB | |
|:---:|:---:|:---:|
| | **Mean (Std. Dev)** | **Count** |
| All | 0.75 (5.33) | 620 |
| 45°N–30°N | −2.32 (5.00) | 91 |
| 10°N–30°N | 0.38 (5.59) | 69 |
| 10°S–10°N | 1.65 (5.15) | 290 |
| 10°S–30°S | 1.64 (5.35) | 155 |
| 30°S–45°S | −1.81 (3.76) | 15 |

**Table 2.** *Cont.*

| SNR (*v/v*) | STAR-ROAB | |
| | Mean (Std. Dev) | Count |
|---|---|---|
| <500 | −1.23 (3.20) | 23 |
| 500–1000 | 0.00 (4.57) | 112 |
| 1000–1500 | 1.19 (5.58) | 179 |
| 1500–2000 | 0.59 (5.56) | 237 |
| >2000 | 1.35 (4.92) | 69 |

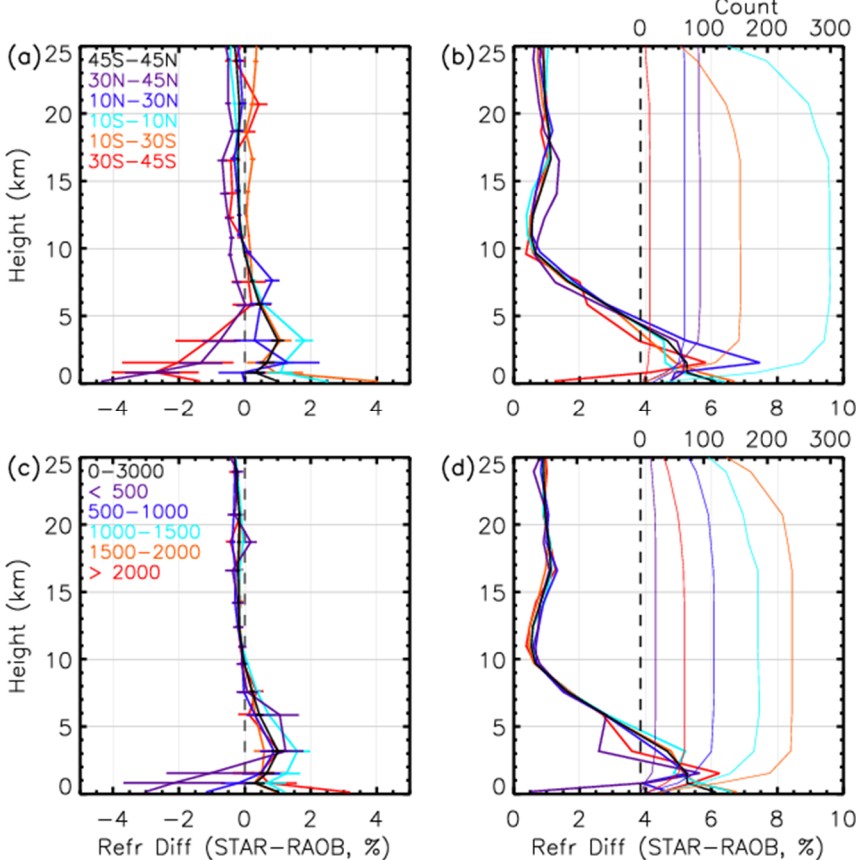

**Figure 12.** Fractional difference of STAR-RAOB (Vaisala RS92) refractivity for RO and radiosonde data collocated between 1–31 October 2019; (**a**) mean, (**b**) standard deviation at different latitude bands, (**c**) mean, and (**d**) standard deviation at different SNR ranges. The count of collocated RO-ROAB pairs is shown by thin lines in (**b**,**d**).

## *5.2. Comparisons to ERA-5*

The bending angle and refractivity retrievals from the STAR inversion package are compared with the ERA-5 bending angle and refractivity interpolated to the reference tangent point locations and time. Figure 13 shows the fractional difference of STAR–ERA5 bending angle and refractivity. The STAR bending angle shows a slightly positive bias in the 2–8 km altitude range and a negative bias below 2 km compared to ERA-5 (see Table 3). Since COSMIC-2 profiles are confined to the tropical and mid-latitude regions, with a greater concentration in the equatorial region (see Figure 3), the mean negative bias is mainly caused by the loss of larger bending angle components in the truncation of the RO signal over tropical regions.

The mean bias and standard deviations of the fractional refractivity difference between STAR and ERA-5 are shown in Table 3. The standard deviations are shown in parenthesis

next to the mean values. The table shows that the mean bending angle biases in height range 0–40 km and 8–40 km are 0.10% and −0.06%, respectively, and the standard deviation is 4.67% and 2.29%, respectively. In the lowest 8 km, the mean bias (standard deviations) at 0–2 km and 2–8 km are −2.39 (18.88)% and 1.70 (12.63)%, respectively.

Compared to UCAR products, STAR products show similar mean biases but relatively more significant standard deviations below 10 km. The larger standard deviation in the STAR retrievals arises from methods used to identifying the bending angle for each impact parameter in the lower troposphere, where the bending angle spectrum is broad due to the presence of multiple frequency components for the same impact parameter. Multipath propagation causes the multiple frequencies to be present in the same impact parameter. For STAR retrievals, we use a sliding mean of a 125-m window centered at a given impact parameter to calculate the bending angle. The bending angle is sensitive to extreme values in the bending angle spectrum within the 125-m window.

The broadening of the bending angle standard deviation in the lower troposphere is mainly caused by measurement and retrieval errors. Measurement errors include uncertainties in the excess phase calculations caused by low SNR, and the retrieval errors include errors caused by the assumption of a spherically symmetric atmosphere. As the RO signals approach the earth's surface, the horizontal variations of water vapor can be significant, causing a broadening of the bending angle spectrum. This broadening of the bending angle spectrum causes larger uncertainties in the retrieved bending angle. Besides, coarser reanalysis data fails to resolve the fine vertical structure of the refractivity variations in the lower troposphere, especially in the boundary layer, which causes larger disparities between RO and reanalysis profiles.

The refractivity biases and standard deviations follow the trend similar to that of bending angle. However, both the biases and the standard deviations for fractional refractivities are smaller, because refractivities are calculated by integrating the bending angle from the top of the atmosphere to the tangent point location. This integration smooths out the variations in the bending angle profiles. The mean and the standard deviations of the refractivity for the 0–40 km altitude are 0.07 and 1.50%, respectively. In the lower troposphere, the mean (standard deviation) of the fractional refractivity difference is equal to −0.36 (4.20)% and 0.37 (2.54)% for the 0–2-km altitude and 2–8-km altitude, respectively.

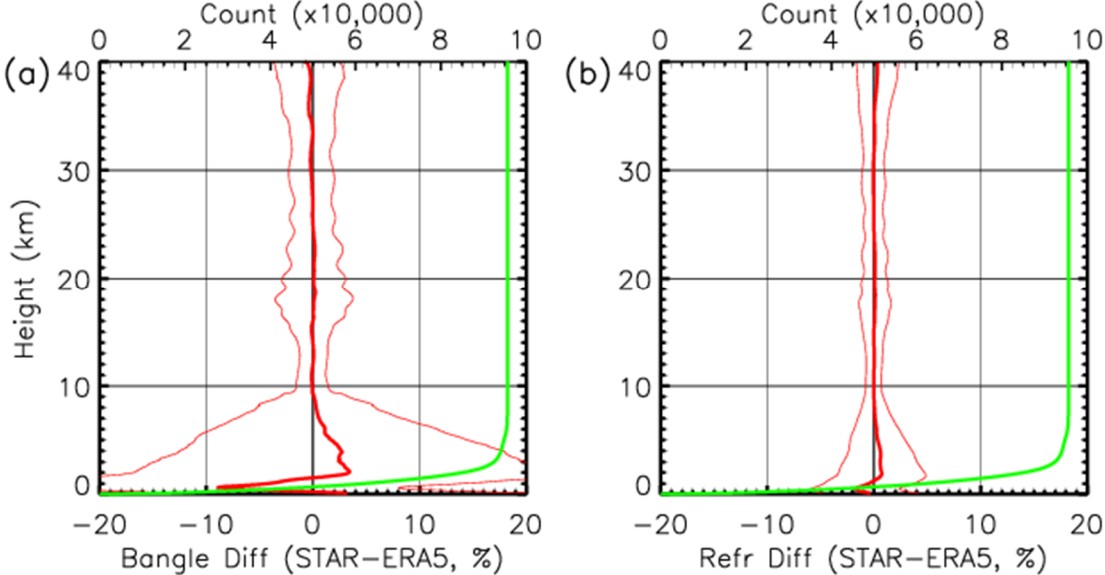

**Figure 13.** Fractional differences of STAR–ERA5 (**a**) bending angles and (**b**) refractivity for data from 1–31 October 2019. The thick red line represents the mean, and the thin lines represent one standard deviation from the mean. Green curves show the number of profiles used for the comparison.

**Table 3.** Bending angle and refractivity fractional difference mean and standard deviation between STAR retrievals with ERA-5 and UCAR retrievals. The standard deviations are given in parenthesis.

| | 0–40 km | | 8–40 km | | 0–2 km | | 2–8 km | |
|---|---|---|---|---|---|---|---|---|
| | ERA5 | UCAR | ERA5 | UCAR | ERA5 | UCAR | ERA5 | UCAR |
| Bending Angle | 0.10 (4.67) | 0.33 (3.56) | −0.06 (2.29) | −0.08 (1.41) | −2.39 (18.88) | 3.73 (15.75) | 1.70 (12.63) | 1.25 (10.74) |
| Refractivity | −0.06 (2.29) | 0.11 (1.19) | 0.05 (1.14) | 0.01 (0.88) | −0.36 (4.20) | 1.14 (3.45) | 0.37 (2.54) | 0.25 (1.97) |

The difference between RO measurements and ERA-5 data has a strong zonal dependence due to the strong zonal variation of water vapor and tropospheric temperature profiles. Figure 14a,b show the mean and standard deviations of the fractional bending angle and refractivity difference between STAR and ERA-5 at different latitude bands from the surface to 5 km. The most significant negative mean biases are at 400–800 m altitude at all latitudes and change from the negative to a positive bias in the altitude range 1–1.5 km.

The 10°–30°N region has the most significant negative bias. This negative bias lies in the tropical and sub-tropical subsidence region with sharp water vapor and temperature change at the marine boundary layer, often associated with super-refraction at the boundary layer interface. The equatorial region between 10°N–10°S also has a significant negative bias due to high moisture variability in the tropics. The smallest biases are in the southern mid-latitude region (30°S–45°S), followed by the northern mid-latitudes (30°N–45°N). The standard deviation minima in the 0–2 km are also observed in the southern mid-latitude region.

The dependence of the fractional bending angle and refractivity biases and standard deviations on the SNR is shown in Figure 14c,d. The biases and standard deviations for the lowest 2 km are shown in Table 4. The negative biases are largest for the low SNR signals (<500 *v/v*), and below 1.5 km, the biases decrease with increasing SNR.

The standard deviations exhibit a trend opposite to the biases from surface to 5 km altitude. The largest standard deviations are observed at the greatest SNR values (>2500 *v/v*) and decrease with SNR. The lowest standard deviation is observed for signals with SNR < 500 *v/v*. The background SNR does not vary based on the SNR of the profiles; however, using a fixed threshold of the background SNR to determine the data truncation point, profiles with the larger SNR values are truncated to lower impact heights than profiles with smaller SNRs. More in-depth analysis is needed to examine this direct relationship between SNR and standard deviation of the fractional bending angles, which will be conducted in a future study.

**Table 4.** The mean and standard deviation (in parenthesis) of the fractional bending angle and refractivity difference between STAR and ERA-5 at different latitude and SNR bands for the 0–2-km height range.

| Latitude Bands | STAR-ERA5 | | SNR (*v/v*) | STAR-ERA5 | |
|---|---|---|---|---|---|
| | Bending Angle | Refractivity | | Bending Angle | Refractivity |
| 45°N–45°S | −2.57 (19.02) | −0.33 (4.19) | <500 | −6.14 (16.20) | −1.10 (3.85) |
| 45°N–30°N | −2.94 (18.66) | −0.49 (4.06) | 500–1000 | −3.56 (17.35) | −0.33 (4.08) |
| 10°N–30°N | −2.67 (18.72) | −0.36 (4.29) | 1000–1500 | −2.54 (19.07) | −0.21 (4.22) |
| 10°S–10°N | −2.57 (18.97) | −0.04 (4.38) | 1500–200 | −2.41 (19.17) | −0.41 (4.16) |
| 10°S–30°S | −2.92 (20.76) | −0.74 (4.28) | 2000–2500 | −1.82 (20.11) | −0.34 (4.29) |
| 30°S–45°S | −1.56 (15.41) | −0.32 (2.97) | >2500 | −1.85 (20.72) | −0.53 (4.33) |

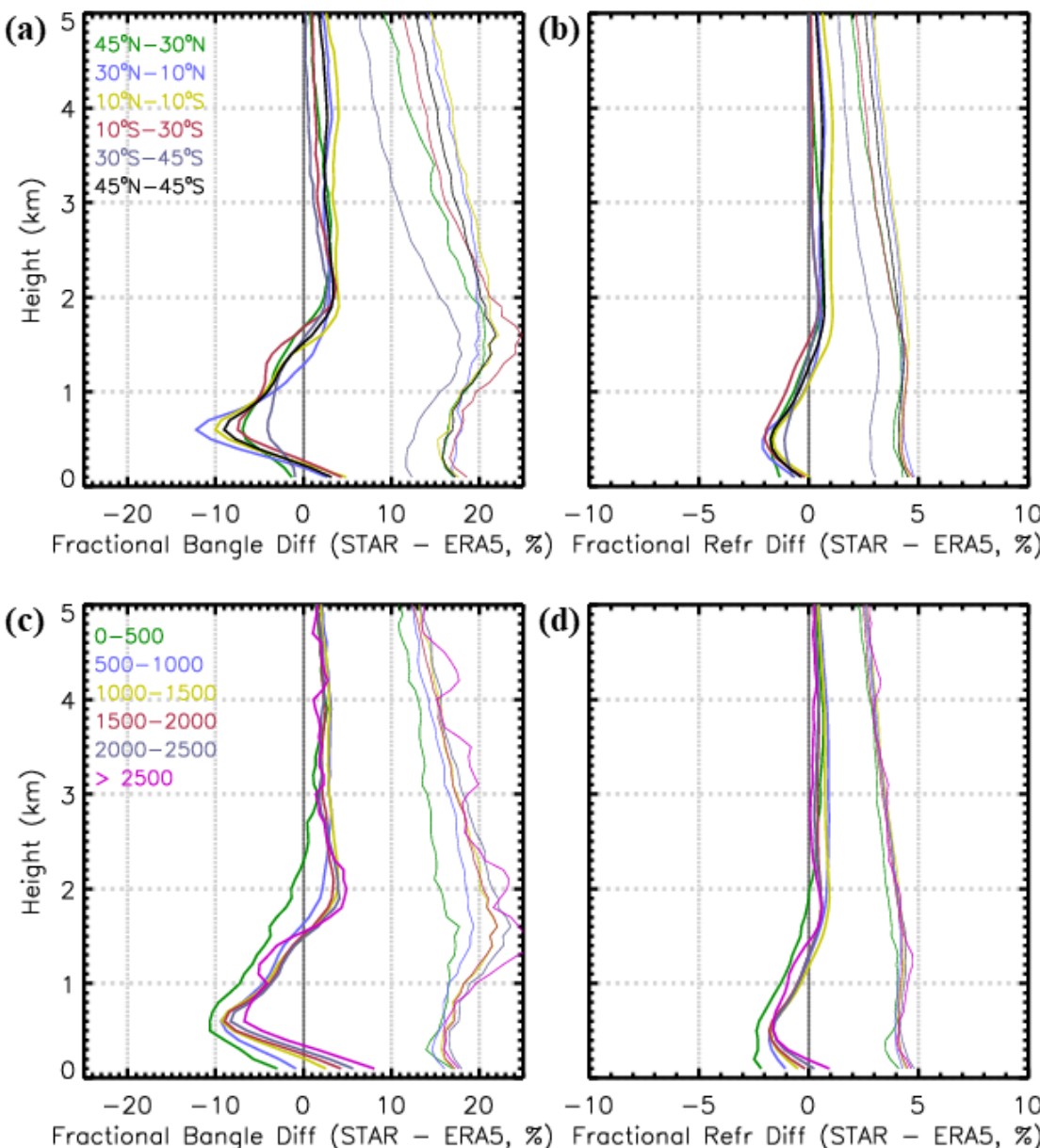

**Figure 14.** Latitudinal distribution of the STAR-ERA5 (**a**) fractional bending angle, (**b**) fractional refractivity, STAR–ERA5 (**c**) fractional bending angle, and (**d**) fractional refractivity at different SNR values. The thick lines represent the mean values, and the corresponding thin lines represent standard deviation.

### 5.3. Comparisons to UCAR Data Products

Comparing the STAR retrievals with UCAR Level 2 bending angle and refractivity are shown as the fractional difference (STAR–UCAR) shown in Figure 15. The standard deviations of the fractional differences are given in Table 3 for 4 different altitude ranges. As shown in Section 5.1, STAR fractional differences compared to ERA-5 show the negative-positive dipole bias in the 0–2-km and 2–8-km altitude ranges. However, the negative bias is larger below 2 km for UCAR than that for STAR. As a result, there is a positive bias in the STAR bending angle of 3.73 and 1.14% in bending angle compared to UCAR. At the 2–8-km range, the positive bias compared to ERA-5 is larger for STAR than UCAR, resulting in a net positive bias of 1.25 and 0.25% for STAR bending angle and refractivity, respectively, compared to UCAR.

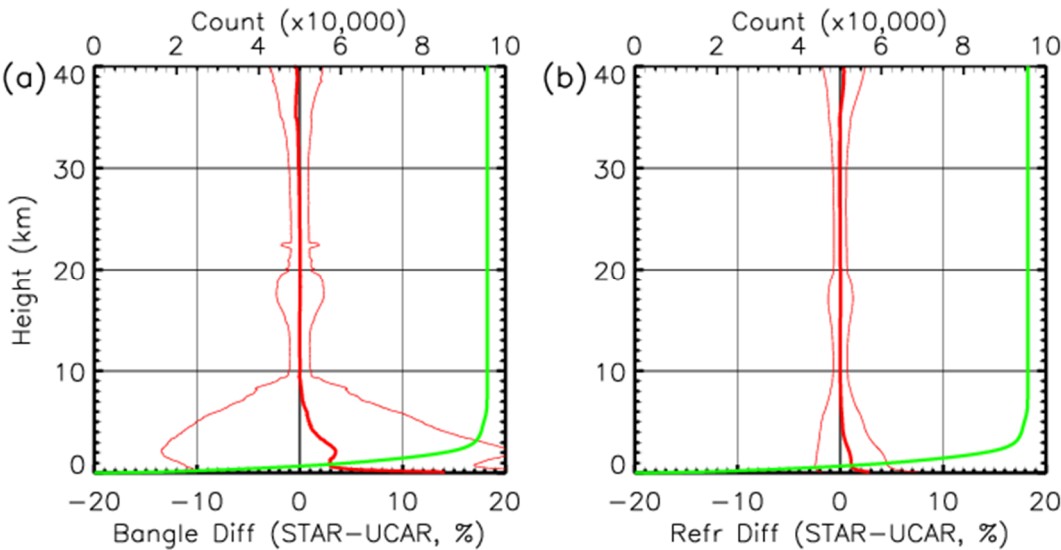

**Figure 15.** (**a**,**b**) Same as Figure 12, but for the STAR–UCAR fractional difference.

Although STAR and UCAR derived RO products use the same L1b data, the methods used at different data processing steps can cause structural differences in the data products, especially in the lower troposphere where the uncertainties are large in the retrieved bending angles. Compared to UCAR below ~8 km, the positive bias of STAR retrievals is mostly due to filtering the raw data used in the excess phase's preprocessing.

Comparing the fractional bending angle and refractivity difference between STAR and UCAR for different latitude and SNR ranges are shown in Figure 16. Comparison between STAR and UCAR shows a consistent positive bias of STAR profile compared in the lower troposphere. This difference is caused by the differences in the ways signal noise is handled in the processing of the excess phases. Both the bias and the standard deviation are smallest in the mid-latitudes in either hemisphere, with the largest bias in the tropics between 10°N–10°S. The dependence on SNR that both the bias and the standard deviations are smallest at low SNR values < 500 *v/v*. Above 500 *v/v*, there is no significant difference in the bias, but the standard deviation shows a weak SNR dependence. The smaller bias and standard deviations at low SNR can be attributed to the weaker penetration depth of low SNR signals in the atmosphere with complex water vapor structure in the lower troposphere. Signals with SNR can penetrate deeper in these complex tropospheric layers with greater horizontal water vapor variations causing larger biases and standard deviations.

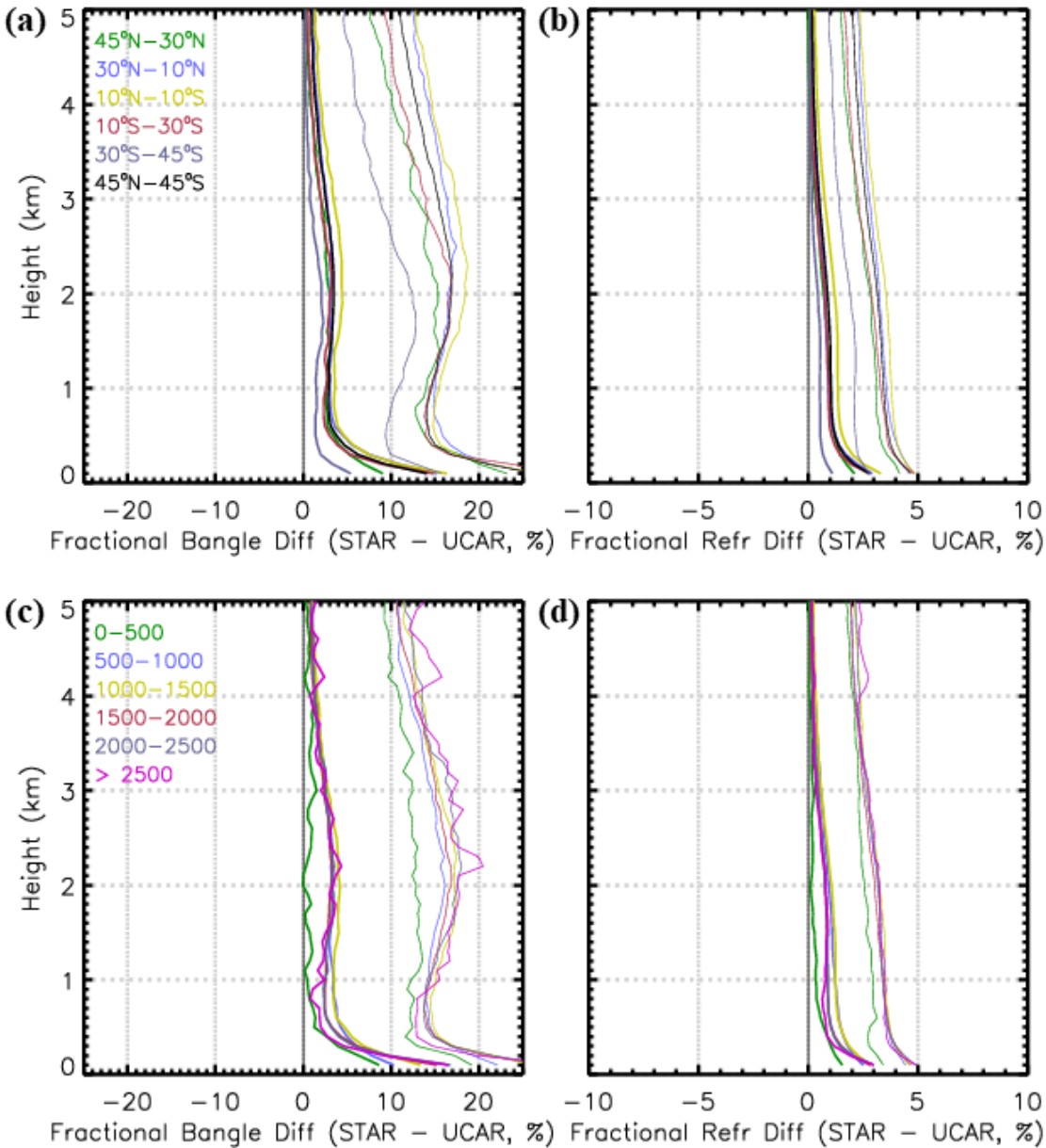

**Figure 16.** (**a–d**) Same as Figure 13, but for the STAR–UCAR fractional difference.

### 5.4. Assessment of Penetration Depth

The penetration depth is the height of the lowest tangent point of the RO profile. It is an important variable in the study of atmospheric processes in the planetary boundary layer. The penetration depth depends on the received signal strength and signal truncation methods used in the RO data processing. RO signals attenuate rapidly as the signals penetrate closer to the earth's surface. For signals where the signal strengths are low, the signal can be lost before reaching the boundary layer. The COSMIC-2 receivers have high gain antennas and have higher SNR than their predecessors, the legacy RO missions. COSMIC-2 measurements are expected to provide better penetration depth (reaching closer to the earth's surface) than the legacy RO missions.

Figure 17 shows the penetration depth obtained from STAR processing. Figure 17a shows the count of the total profiles with penetration depth from 0–3 km at uniform 100 m intervals for October 2019. The total RO profiles normalize the count in that month. For comparison, the penetration depth for UCAR for the same period is also shown. For the October RO profiles, STAR processing provides > 50% profiles with penetration

depth < 300 m, and ~80% of the profiles have penetration depth < 1 km. The penetration depth of the UCAR processed profiles shows even better penetration depth. The penetration depth differences between UCAR and STAR products are due to (1) the differences in how the raw signal is truncated, and (2) the lower limit of the impact parameter is determined after transforming the data is from time to impact parameter space. Figure 17b shows the dependence of penetration depth on SNR for different latitude bands for STAR processed profiles. The figure shows two distinct characteristics: (1) penetration depth increases rapidly below 1000 *v/v* for all latitudes and remain relatively constant for SNR > 1000 *v/v*, and (2) the RO profiles in the mid-latitudes region in both hemispheres (poleward of 30° latitude) penetrate deeper than the profiles in the tropical area.

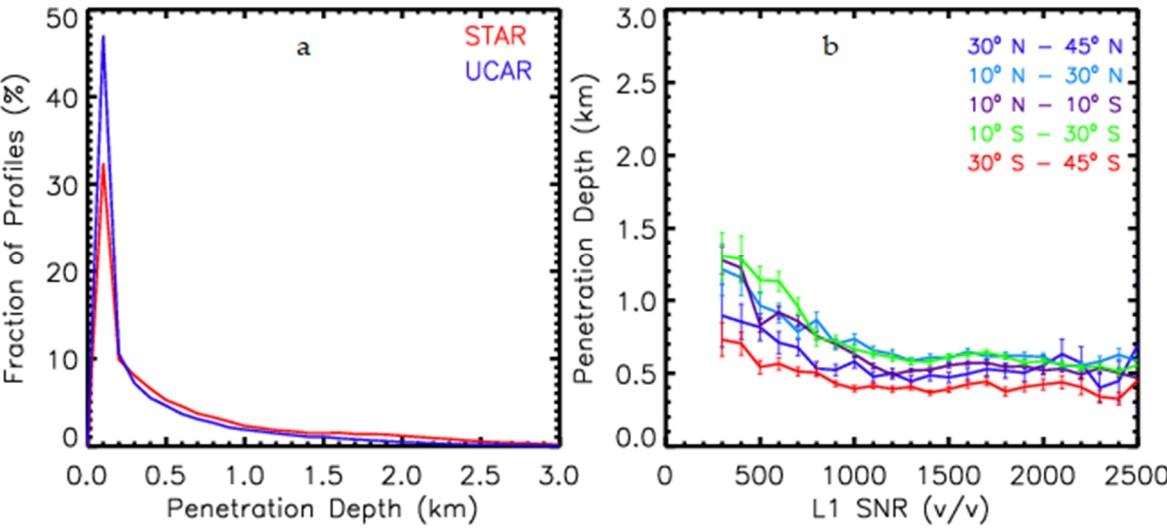

**Figure 17.** (**a**) The probability density function of the penetration depth at 0.1 km height bins, and (**b**) variations of the penetration depth as a function of SNR at 100 *v/v* intervals at different latitude ranges. Vertical bars represent the standard error of the mean.

## 6. Discussions and Conclusions

The initial assessment of the COSMIC-2 data shows that each of the six COSMIC-2 satellites provides more than 900 RO measurements daily and can be expected to provide the anticipated 4000 to 5000 RO profiles daily tropics mid-latitudes. The study finds that the COSMIC-2 position vectors have high stability with the incremental rate of change of the position vectors < 0.1 mm/s for the GNSS positions and <1.4 mm/s COSMIC-2 positions. The SNR, as expected, is twice the SNR of the predecessor COSMIC receivers. The L2 SNR shows dual local maxima at 500 and 1100 *v/v*, corresponding to differences in the L2 SNR of GPS and GLONASS L2 signals.

This study describes the STAR processing system to convert geometry and phase data to bending angle and refractivity profile in detail. In the current version, the data truncation to separate noise from the useful signal is based on empirically determined threshold values above the background SNR, which is calculated for each profile using the highly smoothed SNR of the 10-s of the data at the lowest impact heights. The ad-hoc nature of the SNR threshold method for the data truncation will be improved in the next version of the processing. The initial results indicate that using a single threshold for profiles at all latitudes and SNRs result in inadvertent addition of noisy data in some profiles and higher truncation point in others. It resulted in both larger positive bias in 2–8 km and reduced penetration depth compared to the UCAR profiles.

Compared to the radiosonde measurements, the STAR refractivity has a positive bias in the lower troposphere and a negative bias in the upper troposphere and stratosphere. The bias in the lower troposphere is shown in both zonal dependence and the RO signal strength.

Compared to the reanalysis data, the COSMIC-2 bending angle and refractivity have a negative bias close to the surface (0–1.5 km) and a positive bias from 1.5 km to 8 km. The negative and positive dipole has been analyzed for COSMIC profiles by Sokolovskiy and coauthors [34]. They attributed the biases to the effect of truncation of the signal and the asymmetrical nature of bending angle spectrum caused by noise in the presence of horizontal moisture inhomogeneity. COSMIC-2 profiles also follow the bias pattern observed in the COSMIC profiles. Compared to UCAR profiles, the STAR bending angle and refractivity have a positive bias at below 10 km. This bias is primarily caused by the differences in the ways noise is handled in the STAR processing compared to UCAR processing. The biases from 8–40 km are close to zero (<0.1%) for both UCAR and STAR compared to those computed from ERA-5.

The comparison with ERA-5 shows that biases at the planetary boundary layer are dependent on the latitude and SNR characteristics of the receiver. The biases are largest in the equatorial region between $10°$N and $10°$S and the northern tropics and subtropics region between $10°$N and $30°$N. The smallest biases are in the southern hemisphere mid-latitudes. The smaller bias in the south of mid-latitudes than the northern mid-latitudes is because of the period of the data chosen for the study. During October, northern oceans are warmer and have larger moisture variability than their southern counterparts, causing greater biases in the northern mid-latitudes. The impact of SNR on the RO profiles is complicated and is not completely understood at this point. As expected, the most significant biases are observed at the lowest SNR, and the bias decreases with the increasing SNR. However, the standard deviation shows opposite characteristics, where the standard deviation increases with the increasing SNR.

STAR processing results in more than 50% of the profiles penetrating below 300 m and 80% of the profiles penetrating below 1 km. Overall, the UCAR-processed profiles penetrate deeper than the STAR profiles due to the different methods applied for the data truncation and impact height cutoff after inversion of the bending angle. The penetration depth dependence also shows that it decreases rapidly for SNRs below 1000 *v/v*, but above 1000 *v/v*, there is no significant penetration depth dependence on the SNR.

This paper outlined the independent processing system developed at NOAA STAR and validated the results using ERA-5 and a profile-to-profile comparison with the UCAR-processed profiles. The results show that the current version of the STAR-processed profiles can be used as an independent data source for validation purposes and characterize the structural uncertainties of COSMIC-2 data products.

**Author Contributions:** L.A. and S.-P.H. both contributed to the conceptualization of the research and methodology. X.Z. contributed to the data analysis and investigation. L.A. prepared the first rough draft, but all authors contributed to the final version. All authors have read and agreed to the published version of the manuscript.

**Funding:** This research was funded by NOAA grant NA19NES4320002.

**Data Availability Statement:** Publicly available COSMIC-2 datasets were analyzed in this study. This data can be found here: https://data.cosmic.ucar.edu/gnss-ro/cosmic2/nrt/ (accessed on 5 March 2021) and https://data.cosmic.ucar.edu/gnss-ro/cosmic2/provisional/ (accessed on 5 March 2021).

**Acknowledgments:** The authors would like to thank the anonymous reviewers for their insightful comments and suggestions.

**Conflicts of Interest:** The authors declare no conflict of interest.

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
