# Peer review of "Inverting COSMIC-2 Phase Data to Bending Angle and Refractivity Profiles Using the Full Spectrum Inversion Method"

_remotesensing, doi:10.3390/rs13091793_

Round 1

Reviewer 1 Report

See the attached.

Reviewer 2 Report

The manuscript deals with the calculation banding angle and the reflectivity profile from the COSMIC-2 Phase Data based on the Full Spectrum Inversion Method.

The general comments to the paper are:

  1. In the introduction section, the drawing of the experiment is missing. This will clarify the text.

  1. The applied algorithms are not included in the text; the details are missing. The written description is insufficient. There are only references to the paper with algorithms. This fact generates a lot of questions. Is it possible to put the algorithm to the text with a detailed description?

  1. The scientific contribution of the paper is questionable. Is it an application of the full spectrum inversion method to the data processing of processing or interpretation of the data? Please, clarify.

Line 70: The authors describe multipath propagation. The description is insufficient.

Line 86: Authors wrote, that for the application of the FFT method the circular orbit is needed. This assumption is vague and too strict. The algorithm processes only a short part of the satellite orbit. In this short part, the orbit can be approximated with the circular orbit. The question is the approximation error. Please, clarify. 

Line 216: I line 70 the authors mentioned that the multipath propagation is presented. In line 216 they write, that they use only the carrier phase data and SNR data as an input of the algorithm. The input data completely lose the information about the multipath propagation. Can you clarify this? Is the power delay profile of the channel known? The discussion of the impact of the multipath on the results is missing.

Line 254: The detailed description of the signal truncation is missing. Is it a selection of the relevant part of the signal for further processing? The discussion too long,... too short ... is insufficient.

Line 264: "the final cutoff point is determined by going backward towards lower tangent points where the SNR first drops below 1.5 times the background SNR." Can you precise the SNR determination algorithm? The GNSS receivers measure the C/N0 with some delay.

Figure 4. Unit of SNR. SNR is defined as a ratio of the useful signal power and power of the noise. v is not SI unit.

Variation of the SNR or signal level caused by multipath propagation is called fading.

Lines 277,278 and in many other places: The authors use excess Doppler, Doppler variation, Doppler velocity, and unit m/s, but the unit of Doppler (frequency) is Hz.  The authors should define highlighted physical variables and clarify the text. 

Figure 8.
MDA is not defined. Unit in the y-axis is suspect. The unit of the derivative of the velocity is not usually mm/s.

Round 2

Reviewer 2 Report

Dear editor in chief, dear authors,

all reviewer comments have been considered into text, however, some of them, for instance, comments concerning the processing algorithms in a very minimalistic version. 

A lot of questions concerning SNR and C/N0 are still open. The C/N0 is estimated from despreaded navigation signal and non-correlated noise. The unit is dBc-Hz! SNR value depends on the algorithm's setup! From a radio engineering point of view, SNR is a ratio of the useful signal calculated by some methods of some equivalent noise bandwidth, that is unknown and power of the noise on some, possibly other equivalent noise bandwidth. This data is missing in the paper. 

Despite some small foggy parts, the manuscript is ready to publish.
